# Gold Nanoclusters: Bridging Gold Complexes and Plasmonic Nanoparticles in Photophysical Properties

**DOI:** 10.3390/nano9070933

**Published:** 2019-06-28

**Authors:** Meng Zhou, Chenjie Zeng, Qi Li, Tatsuya Higaki, Rongchao Jin

**Affiliations:** Department of Chemistry, Carnegie Mellon University, Pittsburgh, PA 15213, USA

**Keywords:** gold nanomaterials, electron dynamics, phonon dynamics, optical properties

## Abstract

Recent advances in the determination of crystal structures and studies of optical properties of gold nanoclusters in the size range from tens to hundreds of gold atoms have started to reveal the grand evolution from gold complexes to nanoclusters and further to plasmonic nanoparticles. However, a detailed comparison of their photophysical properties is still lacking. Here, we compared the excited state behaviors of gold complexes, nanolcusters, and plasmonic nanoparticles, as well as small organic molecules by choosing four typical examples including the Au_10_ complex, Au_25_ nanocluster (1 nm metal core), 13 diameter Au nanoparticles, and Rhodamine B. To compare their photophysical behaviors, we performed steady-state absorption, photoluminescence, and femtosecond transient absorption spectroscopic measurements. It was found that gold nanoclusters behave somewhat like small molecules, showing both rapid internal conversion (<1 ps) and long-lived excited state lifetime (about 100 ns). Unlike the nanocluster form in which metal–metal transitions dominate, gold complexes showed significant charge transfer between metal atoms and surface ligands. Plasmonic gold nanoparticles, on the other hand, had electrons being heated and cooled (~100 ps time scale) after photo-excitation, and the relaxation was dominated by electron–electron scattering, electron–phonon coupling, and energy dissipation. In both nanoclusters and plasmonic nanoparticles, one can observe coherent oscillations of the metal core, but with different fundamental origins. Overall, this work provides some benchmarking features for organic dye molecules, organometallic complexes, metal nanoclusters, and plasmonic nanoparticles.

## 1. Introduction

Gold nanomaterials have attracted great interest in both fundamental science and practical applications such as sensing, catalysis, and optoelectronics owing to their unique properties [1,2,3,4,5,6,7,8,9]. For gold nanoparticles with diameters between 3–100 nm, the strong surface plasmon resonance (SPR) dominates in the absorption spectrum, which is caused by collective excitation of free electrons. In contrast, ultra-small gold nanoparticles consisting of tens to hundreds of gold atoms, often called gold nanoclusters, show multiple absorption bands spanning the UV-visible NIR range because of discrete energy levels [10,11]. The significant progress in ligand-protected (e.g., thiolate) gold nanoclusters has allowed atomically precise control of their size and structure [11,12,13]. The structure of a thiolate-protected gold nanocluster typically consists of a metal core with Au–Au bonds (~2.8 Å) and surface Au–S staple motifs [11]. The Au(I) complexes, on the other hand, do not have a metal core, albeit gold aurophilic interactions (Au∙∙∙Au distance > 3 Å) often exist owing to the closed-shell d [10] configuration of Au(I) [14,15]. Typically, solutions of gold complexes are colorless because their absorption peaks lie in the ultraviolet (UV) range (shorter than 400 nm).

Plasmonic gold nanoparticles (AuNPs), gold nanoclusters, and Au(I) complexes have distinct optical features as a result of their differences in size, structure, and bonding. Therefore, understanding the photodynamics will help to deepen the understanding of their electronic structures and optical properties [2,7,9,12,16,17,18]. The electron dynamics of plasmonic gold nanoparticles has been intensively investigated [19,20]. Since the 1990s, El-Sayed and coworkers have probed the size and shape dependent electron dynamics of metallic AuNPs [21,22]. Hartland and Vallee groups have extensively investigated the phonon dynamics and electron–phonon coupling of different sized AuNPs [23,24]. In recent years, there has also been research on the excited state dynamics of ligand-protected gold nanoclusters [1,2,25,26,27]. The electron and phonon dynamics of the nanoclusters were found to be dependent on both size and structure [28,29,30]. Stamplecoskie and Kamat [31] found that the dynamics of Au(I) complexes are different from that of gold nanoclusters. Despite such progress, a systematic comparison of the photo-dynamics between them is still lacking.

Here, we chose Rhodamine B (RB), Au_10_(SR)_10_ complex, Au_25_(SR)_18_ nanocluster, and 13 nm diameter AuNPs protected by citrate (Figure 1) as typical examples to compare the photophysics of small molecules, gold complexes, nanoclusters, and nanoparticles. We employed time-resolved fluorescence and femtosecond transient absorption spectroscopy to probe their excited state behaviors. The electron and phonon dynamics are discussed and compared in detail. The obtained results are of great importance to understand their optical response and further promote their applications in sensing and optics.

## 2. Materials and Methods

Sample preparation: Rhodamine B was purchased from Sigma–Aldrich (St. Louis, MO, USA) and used as received. Au_10_(SR)_10_ (where, SR = 4-tert-butylbenzenethiolate), Au_25_(SR)_18_ (where, SR = 2-phenylethanethiol), and 13 nm diameter Au nanoparticles protected by trisodium citrate were prepared according to the literature [32,33].

Steady state absorption and photoluminescence: UV-vis absorption spectra were measured on a Shimadzu UV-3600plus spectrometer (Kyoto, Japan). Steady-state photoluminescence was measured on a Fluorolog-3 spectrofluorometer from Horiba Jobin Yvon (Piscataway, NJ, USA).

Time-resolved luminescence: fluorescence lifetimes were measured with a time-correlated single photon counting technique from Horiba Jobin Yvon (Piscataway, NJ, USA); a pulsed LED source (376 nm, 1.1 ns) was used as the excitation source.

Transient absorption spectroscopy: Femtosecond transient absorption spectroscopy was carried out using a commercial Ti:Sapphire laser system (SpectraPhysics, 800 nm, 100 fs, 3.5 mJ, 1 kHz) (Santa Clara, CA, USA). A pump pulse was generated using a commercial optical parametric amplifier (LightConversion, Vilnius, Lithuania). A small portion of the laser fundamental was focused into a sapphire plate to produce a supercontinuum in the visible region, which overlapped in time and space with the pump. Multi-wavelength transient spectra were recorded using dual spectrometers (i.e., signal and reference) (Thorlabs, Newton, NJ, USA) equipped with array detectors whose data rates exceed the repetition rate of the laser (1 kHz). Solution samples in 1 mm path length cuvettes were excited by the tunable output of the OPA (pump). Nanosecond transient absorption measurements were conducted using the same ultrafast pump pulses along with an electronically delayed supercontinuum light source with a sub-nanosecond pulse duration (EOS, Ultrafast Systems, Sarasota, FL, USA).

## 3. Results and Discussions

Steady-state absorption and photoluminescence spectra: From the steady-state spectra, one can clearly observe the differences between small molecules and different-sized gold nanomaterials. The RB dye showed significant absorption peaks at ~550 nm and a shoulder at ~520 nm (Figure 2A), which originated from 0–0 and 0–1 vibronic peaks of S_1_ state, respectively. Other absorption peaks at shorter wavelengths were transitions from the ground state to higher excited states than S_1_. The fluorescence spectrum of RB exhibited a mirror image to the S_0_–S_1_ absorption band, with a Stokes shift of 0.1 eV. The Au_10_ complex, on the other hand, showed absorption in the UV region only, with a peak at 346 nm and a shoulder at 380 nm (Figure 2B). The higher energy transitions (*λ* < 300 nm) originated from intraligand (IL) transitions, while the lower energy transitions (346 and 370 nm) should arise from ligand to metal or metal to ligand charge transfer (LM/MLCT) modified by Au(I)–Au(I) interactions [34,35]. The Au_10_ complex showed no observable photoluminescence (PL); however, some of the other gold complexes were reported to exhibit strong PL [36,37,38]. Unlike gold complexes, the Au_25_ nanocluster exhibited multiple absorption bands spanning the entire UV-Vis range (Figure 2C). Theoretical calculations revealed that the absorption band at 670 nm was from the *sp* to *sp* transition while absorption bands at shorter wavelengths involved both *sp* to *sp* and *d* to *sp* transitions [10]. The Au_25_(SR)_18_ exhibited weak photoluminescence (QY < 1%) related to the surface [39]. However, one can observe that the PL peak (centered at 750 nm) overlapped with the lowest absorption band, which indicates that the emission in the visible region may not be the intrinsic PL of Au_25_ nanoclusters. As the size of particles further increased, more and more gold atoms contributed to the electronic states, and finally SPR emerged [8] in the UV-vis spectrum, such as the ~13 nm Au nanoparticles (Figure 2D). Plasmonic gold nanoparticles had a continuous electronic band (i.e., the conduction band), and only had a very weak photoluminescence (QY = 10^−4^) [40]. Below, we further discuss the excited state behaviors of these four species, from which one can find their molecular and plasmonic behaviors.

Organic dyes (Rhodamine B): Before we discuss the photo-dynamics of different sized gold nanomaterials, we first discuss the excited state behavior of organic dyes such as Rhodamine B (RB) for an illustration of molecular behavior. The RB has been widely used as a probe in biological and synthetic polyelectrolyte systems [41]. The excited state behavior of organic dyes has been intensively investigated ever since the birth of time-resolved spectroscopy [42,43,44]. An aqueous solution of RB has strong luminescence (QY = 30%) and the PL lifetime is determined to be 1.7 ns in water (see Appendix A). It is worth noting that the excited state lifetime of RB is highly dependent on solvent polarity (Appendix A), which has been reported previously [45]. In our current work, the transient absorption spectroscopic measurement on RB was performed with excitations at 360 nm and 560 nm, respectively, to excite the sample to the second and first singlet excited state. Upon photo-excitation, one can observe excited state absorption (ESA) around 450 nm, ground state bleaching (GSB), and stimulated emission (SE) between 500 nm and 700 nm (Figure 3A,B). With 360 nm excitation, one can observe an ultrafast decay (~100 fs) in GSB at 520 nm (Figure 3C, blue dip) as well as a rise in SE at 630 nm (Figure 3C, black). On the other hand, the ultrafast decay component was absent with excitation at 560 nm (Figure 3D); thus, the 100 fs component was assigned as internal conversion from S_2_ to S_1_ state. From the kinetic traces, ESA, GSB, and SE decay simultaneously from 1 ps to 3 ns and the transient absorption (TA) lifetime (1.7 ns) matched well with the PL lifetime (Appendix A). The TA spectra and decay dynamics of RB we observed here matched well with the results of previous studies [44,45].

To sum up, the photophysics of RB and other typical organic dye molecules can be well explained by the Jablonski diagram and the relaxation follows the Kasha’s rule [46], that is, the excited state molecules first experience a rapid non-radiative relaxation to the lowest singlet excited state and then emit photons to relax to the ground state.

Gold complexes (Au_10_(SR)_10_): Gold complexes are composed of one or a few gold atoms coordinated by ligands. The Au_10_(SR)_10_ complex is composed of two interlocked Au_5_(SR)_5_ rings and every gold atom is connected to two S atoms [32]. Upon photoexcitation at *λ* = 365 nm, broad positive signals were observed (Figure 4A) with no GSB signal, so the transient signal originates solely from ESA. Such an observation has also been reported in other Au(I) complexes [47]. In the initial 36 ps, the ESA1 around 780 nm decays to give rise to the ESA2 at 535 nm (Figure 4B). In the following 2.8 ns, ESA at all wavelengths decay dramatically by 90%. Decay associated spectra (DAS) obtained by global analysis and singular value decomposition (SVD) of the TA data exhibited three decaying components, 14 ps, 290 ps, and >1 ns (Figure 4C,D). Considering that gold–thiolate complexes show LM/MLCT characteristics in their steady-state UV-vis spectra, photo-excitation at 365 nm can directly generate LM/MLCT excited state [35,37,47]. The first 14 ps process can be assigned to the stabilization and equilibrium of LM/MLCT state. In a previous study, we probed the photodynamics of Au_10_ complexes dissolved in toluene and dichloromethane and found that the decay lifetime was dependent on solvent polarity [48]. Here, we repeated the TA spectra of Au_10_ dissolved in dichloromethane and the same processes can be observed with similar time constants to those in the previous study. Upon photoexcitation, intersystem crossing (ISC) from ^1^LM/MLCT to ^3^LM/MLCT occurred in less than 100 fs, which was not resolved in our TA measurement. The ^3^LM/MLCT state was then stabilized in tens of picoseconds to form a ^3^LM/MLCT^*^ state, which decayed to the ground state (Figure 5). 

Nanoclusters (Au_25_(SR)_18_ as the example): Unlike homoleptic gold complexes, thiolate-protected gold nanoclusters are composed of a well-defined metal core and surface staple motifs (e.g. –S–Au–S–) [11]. The relaxation dynamics of gold nanoclusters of different structures have been investigated by several groups and the relaxation model is more complicated as a result of multiple contributions of both Au and S atoms to the orbitals [27,28,49,50]. Here, Au_25_(SR)_18_ was chosen as an example to illustrate the photophysics. The Au_25_(SR)_18_ structure consisted of an icosahedral Au_13_ core and six Au_2_(SR)_3_ dimeric staple motifs for surface protection (Figure 1) [10]. With excitation at 490 nm, one can observe broad ESA overlapped with GSB peaks at 510 nm, 550 nm, and 675 nm (Figure 6A), which is a typical feature of gold nanoclusters [2,50,51]. During the first picosecond, one can observe a broad ESA band between 500 nm and 620 nm decaying rapidly and the time constant was ~600 fs (Figure 6B). The rapid relaxation was not observed under excitation of 800 nm (Figure 6C), which suggests that it should be internal conversion from higher to lower excited states. With excitation at 800 nm, the TA spectrum at ~0.3 ps was equal to the spectrum with excitation at 490 nm after 2 ps, which indicates that the 800 nm pulse excited Au_25_ directly to the lower excited state. The TA signal did not decay significantly between 2 ps and 3 ns, indicating a significantly longer excited state lifetime of Au_25_ than 3 ns. It was also interesting to see that the nanosecond TA and time resolved-PL give different lifetimes (Appendix A). Compared with the dynamics of the Au_10_ complex, the excited state dynamics of the Au_25_ nanoclusters behaved more like that of small molecules.

With excitation at 800 nm, one can also observe coherent oscillations in the first few picoseconds (Figure 6C,D). Two frequencies (40 cm^−1^ and 80 cm^−1^) were exhibited at different probe wavelengths (Figure 6D), which have been reported previously [51]. These oscillations observed in TA decays, also observed in other nanoclusters [52,53], were assigned to acoustic vibrations of the metal core and explained as displacive excitation [52,54], similar to that observed in semiconductors and small molecules [54,55].

Plasmonic nanoparticles (13 nm diameter AuNPs): As the particle size becomes larger, more gold atoms will contribute to the electronic states, and eventually the bandgap will disappear, giving rise to a transition from non-metallic to metallic [8]. The TA spectra of 13 nm AuNPs showed a significant bleaching signal at 520 nm and ESA on two sides (Figure 7A). We found that (i) the GSB became sharper from 0.2 ps to 5 ps and (ii) the higher pump fluence gave rise to a broader bleaching signal compared to that of the lower pump fluence (Figure 7A,B). The broadening of GSB in the initial time delay (full width at half maximum decreased from 50 nm to 30 nm in Figure 7A) is ascribed to the heating effect [20]. Upon photo-excitation, electrons in the AuNPs will be heated to a very high temperature (e.g., 1000 K, depending on the pump power). Heating the nanoparticle will result in broadening of the SPR peak as well as the GSB in the TA spectra. After the electrons are heated to a very high temperature, the excited state energy will first reach equilibrium via electron–electron scattering (manifested in the 100 fs rise of the kinetic traces in Figure 7C). Subsequently, the energy will be transferred from the electrons to the lattice through electron–phonon coupling (1–5 ps rapid decay in Figure 7C) [3,24]. Accompanied by the rapid decay, the GSB will also be narrowed during the electron–phonon coupling process (Figure 7A,B). Finally, the energy will dissipate into the environment by phonon–phonon relaxation, which is dependent on the surrounding medium (100 ps decay in Figure 7C). As there is no bandgap in plasmonic Au nanoparticles, there is no electron–hole separation or recombination process as in nanoclusters. Instead, the relaxation dynamics can be described by a well-established two-temperature model [21,56]. With the high pump fluence, one can observe a prominent oscillation with a frequency of 7.5 cm^−1^ (4 ps in periods) in the 13 nm AuNPs (Figure 7D). Unlike that of the gold nanoclusters, the oscillations in the metallic AuNPs originated from the periodic shift of the SPR band due to the lattice expansion induced by laser heating [20]. In addition, the oscillations in the metallic NPs can be well modelled by a classical continuum model (scaling as 1/*D*, where *D* is the diameter), but this model fails in gold nanoclusters. 

A distinct feature of metallic gold nanoparticles is that the electron–phonon coupling is dependent on pump fluence [3,19]. When the pump fluence increased from 80 to 1800 uJ/cm^2^, one can clearly observe that the electron–phonon coupling slowed down significantly (Figure 8A). After plotting the fitted time constants as a function of laser fluence, a linear relationship can be observed (Figure 8B), which agrees well with previous studies [3,56]. Such a power dependence is only observed in plasmonic nanoparticles, which can be well explained by the two-temperature model [21]. Compared to the plasmonic NPs, the TA measurements of RB, Au_10_ complex, and Au_25_ nanoclusters under different pump fluences exhibited no differences in the decay dynamics (Appendix A), hence, they were power independent. This serves as a signature of the non-metallic state. In Table 1, the photophysical features of the four types of materials are summarized. In Scheme 1, the relaxation processes as well as the time constants are illustrated.

## 4. Conclusions

In summary, we have compared the photophysical properties of small organic molecules, gold complexes (no explicit core), nanoclusters (with a core and surface Au–S staples), and metallic-state nanoparticles by choosing four examples (Rhodamine B, Au_10_(SR)_10_, Au_25_(SR)_18_, and 13 nm diameter AuNPs). Femtosecond transient absorption spectroscopy was used to determine their relaxation time constants and photodynamics. Overall, gold nanoclusters behave similarly to small molecules, for example, showing a rapid internal conversion (<1 ps) and a long-lived excited state lifetime (~100 ns). In Au(I) complexes, LM/MLCT charge transfer states dominate the relaxation dynamics and no sub-picosecond relaxation was observed. On the other hand, the electron dynamics of plasmonic gold nanoparticles can be well explained by a two-temperature model, with no electron–hole separation or recombination being observed. Overall, the revealed features in photodynamics of the four different materials provide some benchmarking features and are expected to be of great importance for understanding their electronic structures and broadening their applications in various fields in future work.

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
