# Peer review of "Gold Nanoclusters: Bridging Gold Complexes and Plasmonic Nanoparticles in Photophysical Properties"

_nanomaterials, 2019, doi:10.3390/nano9070933_

Reviewer 1 Report

This is a well-written and informative study on the photophysical behavior of well-defined gold clusters of varying sizes as compared to small molecules. The methods seem to be sound and the analysis of the data is easy to follow. There are two minor points that I would make:

There are comments made in the Introduction and Conclusion about the usefulness of this study in characterizing different properties of the gold clusters and nanoparticles for various applications. Are there any specific applications that the authors could provide where one size or another would be preferred?

 In discussing the properties of the Au25 clusters, it is noted that they behave more like a small molecule than the Au10 clusters do. Is there a reason that this might be the case? It seems backward that the larger cluster should behave more like a small molecule.

Overall this study should be of significant interest to the nanocluster community, and should be published.

Author Response

Dear reviewer,

Thanks for your valuable comments,
please see the attached files.

Reviewer 2 Report

This article is concerning with the photodynamics of several Au compounds: Au complex (Au10), Au nanocluster (Au25), and Au nanoparticle (13 nm). The results can be summarized as below:
- Transient absorption spectra of Au complex, Au nanocluster, and Au nanoparticle were analyzed and compared with a fluorescent organic dye (rhodamine B: RhB).
- As a standard or a control, RhB demonstrated a rapid process (< 100 fs: relaxation from S2 to S1) and a slow process (~ several nano-seconds: S1 to G).
- Au complex (Au10) exhibited two excited state absorption (ESA) at 780 and 535 nm. The relaxation process was analyzed to be three components (14 ps to form 3LM/ML-CT*, 290 ps and nano-second to G). The rapid process (< 100 fs) for excitation and ISC was not measured because of the resolution limitation.
- Au nanocluster (Au25) showed a rapid process (< 600 fs: corresponding to the relaxation from S2 to S1 in RhB) and a slow process (> 3 nano-seconds). In the rapid process, ESA (600-620 nm) decreased and overlapped with GSB peaks (510, 550, and 675 nm).
- AuNP indicated a sharp GSB (520 nm) overlapping with a broad ESA (400-800 nm). The process consisted of three phases: electron-electron scattering (< 100 fs), electron-phonon coupling (1-5 ps), and energy dissipation (~100 ps).
- Au25 and Au nanoparticle exhibited oscillations (40-80 cm-1 for Au25 and 7.5 cm-1 for Au nanoparticle), which were assigned to the acoustic vibration of Au lattice.
- The electron-phonon coupling in Au nanoparticle showed a linear correlation with the excitation energy, which was not observed in the others.
In my opinion, the aim of this article is good. Photodynamics of Au nanostructures should be clearly understood to design devices in various applications, and this article will help it. Results are well shown, and the differences between the nanomaterials are easily understood.
However, I found some exaggeration and misleading, which should be considered before publication. I listed them below. I wish my comments will help to improve the manuscript.
(1) Discussion
- In abstract, Line 179, and Line 240, author insists that “gold nanoclusters behave like small molecules”. However, their behaviors were not so similar. In the RhB, the rapid process (S2 to S1) resulted in the spikes in GSB and SE, but not in the ESA (Figure 3C). On the other hand, the Au25 demonstrated the rapid decline of GSB and ESA (Figure 6B). The ESA was constant in RhB and change in Au25.
Moreover, the rapid process (< 100 fs) in Au10 was not measured, and the comparison by 2 excitation wavelengths was not examined. The RhB and Au25 were excited by 2 excitation waves to confirm the different energy states. Therefore, the similarity in RhB, Au10, and Au25 should not be concluded in this time. (It depends on the definition of “similar”)
In my opinion, it is not necessary to say about the similarity. A systematic illustration (e.g. timing diagram) of relaxation processes in 4 compounds can be enough, for instance,

RhB:         S1 → S2                        → G
Au10:1LM/ML-CT → 3LM/ML-CT → 3LM/ML-CT* → G
Au25: higher excited state → lower excited state   → G
AuNP: electron-electron    → phonon     → dissipation
Time scale:   < 100 fs    < 1 ps     < 10 ps   <100 ps   < 1 ns -----
(Sorry, too rough)

- Line 195-197: the “sharp” and “broadening” are not clearly confirmed. Please use an index, such as FWHM to define the broadness. Additionally, the peak position of GSB blue-sifted (Figure 7A and B). It could support the explanation of mechanism in GSB decreasing.
(2) Minor points
- Figure 1: Please explain the meaning of colors (blue and pink)
- Figure 1 caption “Atomic structure”: Is molecular structure better?
- Undefined abbreviations: Line 129 “TA” for transient absorption and Line 156 “DCM” for dichloromethane.
- Line 133: text size.
- Figure 7D is nor referred in the text (should be in Line 210).

Author Response

Dear reviewer,

Thanks for your valuable comments,
please see the attached files.

Round  2

Reviewer 2 Report

The author addressed the questions offered in the round 1, and the manuscript was sufficiently corrected.

In my opinion, the manuscript can be published after minor revisions.

e.g.

Line 160 "TA" can be used after Line 135

Line 159 "DCM", instead of Line 173 etc.

Author Response

We thank the reviewer. Revisions are as follows,

1) We have replaced "transient absorption" with "TA" in lines 153 and 159.

2) We have removed the abbreviation of DCM since it's used in two places only.